



**Interactions between aerosol organic components and liquid water content**
**during haze episodes in Beijing**
Xiaoxiao Li[1], Shaojie Song[2], Wei Zhou[1], Jiming Hao[1], Douglas R. Worsnop[3,4], and Jingkun Jiang[1]*
[1]State Key Joint Laboratory of Environment Simulation and Pollution Control, School of Environment, Tsinghua University,
Beijing, 100084, China
[2]School of Engineering and Applied Sciences, Harvard University, Cambridge, Massachusetts 02138, USA
[3]Institute for Atmospheric and Earth System Research / Physics, Faculty of Science, University of Helsinki, Finland
[4]Aerodyne Research Inc., Billerica, Massachusetts 01821, USA
*: *Correspondence to*: J. Jiang (jiangjk@tsinghua.edu.cn)
**Abstract:** Aerosol liquid water (ALW) is ubiquitous in ambient aerosol and plays an important role in the formation of both
aerosol organics and inorganics. To investigate the interactions between ALW and aerosol organics during haze formation and
evolution, ALW was modelled based on long-term measurement of submicron aerosol composition in different seasons in
Beijing. ALW contributed by aerosol inorganics (ALW$_{inorg}$) was modelled by ISORROPIA-II, and ALW contributed by
organics (ALW$_{org}$) was estimated with κ-Köhler theory, where real-time hygroscopicity parameter of the organics ($\kappa_{org}$) was
calculated from the real-time organic oxygen-to-carbon (O/C). Overall particle hygroscopicity ($\kappa_{total}$) was computed by
weighting component hygroscopicity parameters based on their volume fractions in the mixture. We found that ALW$_{org}$, which
is often neglected in traditional ALW modelling, contributes a significant fraction (18-32%) to the total ALW in Beijing. The
ALW$_{org}$ fraction is largest in the cleanest days when both the organic fraction and $\kappa_{org}$ are relatively high. The large variation
of O/C, from 0.2 to 1.3, indicates the wide variety of organic components. This emphasizes the necessity of using real-time
$\kappa_{org}$, instead fixed $\kappa_{org}$, to calculate ALW$_{org}$ in Beijing. The significant variation of $\kappa_{org}$ (calculated from O/C), together with
highly variable organic or inorganic volume fractions, leads to a wide range of $\kappa_{total}$ (between 0.20 and 0.45), which has great
impact on water uptake. The variation of organic O/C, or derived $\kappa_{org}$, was found to be influenced by T, ALW, and aerosol mass
concentrations. Among which, T and ALW both have promoting effects on O/C. During high-ALW haze episodes, although
the organic fraction decreases rapidly, O/C, and derived $\kappa_{org}$, increase with the increase in ALW, suggesting the formation of
more soluble organics via aqueous/heterogeneous-phase process. A positive feedback loop is thus formed: during high-ALW
episodes, increasing $\kappa_{org}$, together with decreasing particle organic fraction (or increasing particle inorganic fraction), increases
$\kappa_{total}$, thus further promotes the ability of particles to uptake water.

**1 INTRODUCTION**
Aerosol liquid water (ALW) is a ubiquitous component of ambient aerosol and exerts great influences on aerosol physical and
chemical properties, especially in regions with high relative humidity (RH) (Cheng et al., 2016; Cheng et al., 2008; Covert et



al., 1972; Ervens et al., 2014; Nguyen et al., 2016; Pilinis et al., 1989; Zheng et al., 2015). From the perspective of aerosol
physical processes, ALW influences particle lifetime, optical properties, radiative forcing, and the ability of particles to deposit
in the humid human respiratory tract (Andreae and Rosenfeld, 2008; Cheng et al., 2008; Covert et al., 1972; Löndahl et al.,
2008). ALW also promotes partitioning of some of the inorganic gases and water-soluble organic gases to the condensed phase,
thus directly increasing aerosol mass loadings (Asa-Awuku et al., 2010; Parikh et al., 2011). From the perspective of aerosol
chemical processes, ALW can serve as a reactor for heterogeneous/aqueous reactions, facilitating the formation of both
secondary inorganics (Cheng et al., 2016; Sievering et al., 1991; Wang et al., 2016) and organics (Carlton et al., 2009; Ervens
et al., 2014; Song et al., 2019). As a result, understanding ALW content is critical in clarifying the formation and evolution of
ambient aerosols as well as their impacts on air quality and climate, especially in urban cities like Beijing where severe haze
events take place frequently with elevated RH (Sun et al., 2013; Zheng et al., 2015).

The interaction between ALW and aerosol chemical composition is a key issue for haze formation but remains uncertain,
especially regarding the interaction between ALW and aerosol organics. Studies have demonstrated that secondary inorganic
aerosol (SIA) and secondary organic aerosol (SOA) surpass primary species during haze formation in China (Huang et al.,
2014; Sun et al., 2016; Zheng et al., 2016). SOA or SIA-driven haze formation is widely observed to be associated with elevated
relative humidity (RH), especially in winter. In Beijing, as RH rising from below 40% to above 60%, the following has been
reported: (1) aerosol mass loadings increase significantly; (2) particles phase transition from solid/semisolid to liquid phase
(Liu et al., 2017); (3) sulfur and nitrogen oxidation ratios both increase (Cheng et al., 2016; Sun et al., 2013; Zheng et al.,
2015); (4) water-soluble inorganics increase faster than organics (Liu et al., 2015; Quan et al., 2015; Sun et al., 2013; Zheng
et al., 2015). RH affects secondary species via heterogeneous/aqueous phase uptake or reactions. During haze episodes, gas
phase photochemical formation of SIA and SOA is largely suppressed by the weakened solar radiation (Zheng et al., 2015).
Formation of SIA and SOA is thus suggested to be dominated by heterogeneous/aqueous phase reactions (Xu et al., 2017),
which are largely dependent on ALW. Based on ALW measurements, previous studies have proposed positive feedback loops
in which elevated RH increases particle concentration and particle inorganic fraction; increased particle concentration and
inorganic fraction in turn increase the water uptake (Cheng et al., 2016; Liu et al., 2017; Wu et al., 2018). However, whether
or how elevated ALW affects the evolution of SOA during haze episodes remains less understood than that of SIA because of
the complexity of SOA species.

Long-term data are needed to evaluate the amount of ALW and its interactions with aerosol organic compositions. So far,
short-term ALW data in Beijing (Bian et al., 2014; Fajardo et al., 2016) have been collected by directly measuring size-resolved
aerosol hygroscopic volume growth factors (VGF) and particle size distributions by hygroscopicity-tandem differential
mobility analyzer (H-TDMA) (Rader and McMurry, 1986) or dry-ambient aerosol size spectrometer (DAASS) (Engelhart et



al., 2011; Stanier et al., 2004). However, long-term measurements of ALW are rare because of the challenge in sustaining these
instruments. Another approach to obtain ALW is to combine aerosol chemical composition measurements and model
predictions. ALW contributed by inorganics can be modelled by inorganic thermodynamic equilibrium models, such as
ISORROPIA-II (Fountoukis and Nenes, 2007; Nenes et al., 1999, 1998), E-AIM (Clegg and Pitzer, 1992; Clegg et al., 1992),
and SCAPE II (Kim et al., 1993a, b). Modelled inorganic water content is usually regarded as the total ALW because inorganic
salts contribute a large fraction of the total particle loading and the hygroscopicity of inorganic salts is much larger (~6 times)
than those of organic species (Bian et al., 2014; Hennigan et al., 2008). Although this approximation provides reasonable ALW
in many ambient conditions, it fails in some cases. Especially when organics contribute a dominant fraction to particle loading,
large discrepancies arise between the modelled inorganic water and the actual ALW content (Fajardo et al., 2016). Therefore,
it is important to take the organic contribution to ALW into consideration. specific models include the calculation of organic
ALW; e.g., aerosol diameter dependent equilibrium model (ADDEM) (Topping et al., 2005b, a). However, application of such
models is hindered by lack of long-term measurements of specific OA species.

Recent studies have proposed a method to predict total ALW using the non-refractory submicron particulate matter (NR-PM$_1$,
particle diameter between 40 nm and 1 μm) composition measured with the widely used Aerosol Mass Spectrometer (AMS)
(Nguyen et al., 2014; Nguyen et al., 2016). The inorganic contribution to ALW (ALW$_{inorg}$) was modelled by ISORROPIA-II;
organic contribution to ALW (ALW$_{org}$) was estimated with κ-Köhler theory (Petters and Kreidenweis, 2007; Su et al., 2010).
The total aerosol liquid water (ALW) is then the sum of ALW$_{inorg}$ and ALW$_{org}$. ALW estimated by this method, which only
requires aerosol chemical composition obtained from AMS measurements (Zhang et al., 2007), corresponds reasonably with
measured ALW. Thus, this method can be used to predict long-term ALW from aerosol chemical composition and to explore
interactions between ALW and organic evolution during haze events.

In this study, long-term NR-PM$_1$ chemical composition measurement was used to predict ALW in Beijing during various
seasons (292 days in 5 years). ALW$_{org}$ and ALW$_{inorg}$ were estimated using κ-Köhler theory and ISORROPIA-II, respectively.
A real-time organic hygroscopic parameter ($\kappa_{org}$, calculated from organic O/C ratio) was used to estimate ALW$_{org}$. The
relationship between the total ALW and $\kappa_{org}$ was explored. Within this long-term dataset, 12 high-ALW haze episodes and 8
low-ALW haze episodes were identified. Chemical evolution during high-ALW and low-ALW haze episodes was found to
differ significantly. Positive feedback between organic hygroscopicity, organic volume fraction, overall particle hygroscopicity,
and ALW is proposed to be a factor driving severe haze formation in Beijing during high-ALW episodes





## 2 METHODOLOGY


### 2.1 Long-term measurements of particle chemical composition


Long-term field measurements were carried out between December 2013 and August 2017 at an urban site located on the
campus of Tsinghua University in Beijing. The monitoring site is located on the top floor of a four-storey building without
other tall buildings nearby with detailed information provided elsewhere (Cai and Jiang, 2017; Cao et al., 2014; He et al.,
2001). Data from 292 days were used, including 2-3 months' data from each of the four seasons (Table S1). The average NR-
$PM_1$ mass concentrations from spring to winter were 81.1, 54.2, 63.9, and 63.2 μg m$^{-3}$, respectively. Note that $PM_{2.5}$
concentrations in Beijing were decreasing during this period (http://www.bjepb.gov.cn/).

Chemical composition of NR-$PM_1$, including sulfate ($SO_4^{2-}$), nitrate ($NO_3^-$), ammonium ($NH_4^+$), chloride ($Cl^-$), and total
organics (Org), was measured using a quadrupole aerosol chemical speciation mass spectrometer (Q-ACSM)(Ng et al., 2011).
The Q-ACSM was calibrated before each measurement following the procedure described by Ng et al., (2011). The
meteorological conditions, including temperature (*T*), relative humidity (RH), and other routine meteorological parameters,
were recorded by a meteorological station.

### 2.2 Aerosol liquid water modelling


$ALW_{inorg}$ was modelled by ISORROPIA-II using meteorological conditions and the Q-ACSM measured inorganic
compositions. The model was carried out with "reverse" and "metastable" mode. Compared to the "stable" mode, "metastable"
mode assumes that particles are always aqueous droplets, even at low RH. Although some earlier studies observed phase
transitions of ambient particles, recent studies suggest that ambient aerosols tend to be in "metastable" states due to the
coexistence of organic compounds that inhibit or cover up the deliquescence and efflorescence behavior of inorganic
compounds (Martin et al., 2008; Rood et al., 1989). The "metastable" mode predicts more water than predicted from "stable"
mode when RH is between 40% and 70%, while similar with the latter when RH is above 70% or below 40% (Fig. S1),
consistent with previous work (Song et al., 2018). In a few of the modelling results in summer and autumn, high acid/base
ratio caused some of the $NO_3^-$ and $Cl^-$ to enter the gas phase in the form of $HNO_3$ and HCl, resulting in disagreement between
the output liquid phase $NO_3^-$ and $Cl^-$ and the input aerosol phase $NO_3^-$ and $Cl^-$. These points were removed.

$ALW_{org}$ was estimated using a simplified equation of κ-Köhler theory where Kelvin effect was neglected (Petters and
Kreidenweis, 2007) (Eq. 1),
$$ALW_{org} = V_{org}\kappa_{org}\frac{a_w}{1-a_w} \qquad (1)$$





where $a_w$ is the water activity and was assumed to be the same as RH (Bassett and Seinfeld, 1983) and $V_{org}$ is the volume
concentration of organics measured by Q-ACSM (density of organics was assumed to be 1.2 g cm$^{-3}$). In previous studies, a
fixed $\kappa_{org}$ in the range of 0.06-0.13 was used for urban, urban downwind, and rural sites (Gunthe et al., 2011; Nguyen et al.,
2016; Rose et al., 2011). However, the hygroscopicity of organics is highly variable and $\kappa_{org}$ can vary between 0 and 0.3 for
different species (Lambe et al., 2011; Massoli et al., 2010). $\kappa_{org}$ was found to have a positive linear relationship with organic
O/C ratio (Chang et al., 2010; Dick et al., 2000; Duplissy et al., 2011; Gunthe et al., 2011; Petters et al., 2009), which likely
reflects combined effects of molecular weight, volatility, and surface activity (Nakao, 2017; Wang et al., 2019). Previous
studies proposed several empirical methods to calculate $\kappa_{org}$ from O/C derived from a series of chamber and field experiments
(Chang et al., 2010; Duplissy et al., 2011; Jimenez et al., 2009; Lambe et al., 2011; Massoli et al., 2010). Comparing these
methods (Table S2), Eq. 2 was used to calculate real-time $\kappa_{org}$ over a broadest O/C range (0.05-1.42) (Lambe et al., 2011),
$$\kappa_{org} = (0.18 \pm 0.04) \times O/C + 0.03 \qquad (2)$$
where real-time O/C was calculated from Q-ACSM measured $f_{44}$ (the fraction of $m/z$ 44 fragments signal to total organic signal,
O/C = 0.079 + 4.31 × $f_{44}$) which has been widely used to study the aging process of OA species (Canagaratna et al., 2015; Ng
et al., 2010).

The Zdanovskii-Stokes-Robinson (ZSR) mixing rule was used to calculate the total ALW. According to ZSR, the total water
uptake into internally mixed particles is the sum of water content uptake by each pure component (Jing et al., 2018).

Particle hygroscopic volume growth factor (VGF) is the ratio of the volume of the wet particle to the corresponding particle
volume at dry conditions. The size-independent VGF was calculated using Eq. 3,
$$\text{VGF} = \frac{\sum \frac{m_{i,ACSM}}{\rho_i} + (ALW_{inorg} + ALW_{org})/\rho_{water}}{\sum \frac{m_{i,ACSM}}{\rho_i}} \qquad (3)$$
where $m_{i,\,ACSM}$ is the mass concentration of species "i" measured by Q-ACSM. The densities were assumed to be 1.75, 1.75,
1.75, 1.52, 1.2, and 1.0 g cm$^{-3}$ for sulfate, nitrate, ammonium, chloride, organics, and water, respectively (Salcedo et al., 2006).

Overall particle hygroscopicity ($\kappa_{total}$) was computed by weighting component hygroscopicity parameters by their volume
fractions in the mixture (Dusek et al., 2010; Gunthe et al., 2009; Petters and Kreidenweis, 2007) (Eq. 4),
$$\kappa_{total} = \kappa_{inorg} \cdot frac_{inorg} + \kappa_{org} \cdot frac_{org} \qquad (4)$$
where $frac_{inorg}$ and $frac_{org}$ are the inorganic and organic volume fractions in NR-PM$_1$, respectively. Inorganic species are mainly
in the form of NH$_4$NO$_3$, H$_2$SO$_4$, NH$_4$HSO$_4$, and (NH$_4$)$_2$SO$_4$ (Liu et al., 2014); corresponding hygroscopic parameters were
0.68, 0.68, 0.56, and 0.52, respectively. As a result, an average value of 0.6 was used as the hygroscopicity parameter of the
inorganic components ($\kappa_{inorg}$), with the assumption that the relative abundance of NH$_4$NO$_3$, H$_2$SO$_4$, NH$_4$HSO$_4$, and (NH$_4$)$_2$SO$_4$





does not change significantly. Thus in our study, variation of $\kappa_{total}$ with RH only reflects changes in $frac_{org}$ and $\kappa_{org}$.
**2.3 Haze episode identification**
The haze pollution in Beijing have shown typical evolution pattern where a pollution episode usually starts with a clean day,
then accumulates for 2-7 days, and eventually disappears within 1-2 days (Guo et al., 2014; Zheng et al., 2016). In this study,
22 haze episodes were identified (Table S3). Only episodes containing 4 or more than 4 calendar days were taken into
consideration. The haze episodes were further classified according to ALW volume fraction; that is, the ratio of ALW volume
to the wet particle total volume (ALW volume fraction = $V_{ALW} / (V_{ALW} + V_{NR-PM1})$). 12 were distinguished as high-ALW haze
episodes (ALW volume fraction > 0.3 for at least 50% of the haze period), while 8 were distinguished as low-ALW haze
episodes. All 20 distinguished episodes were associated with growing RH, the other 2 two with irregular RH variations were
classified as undefined. Average NR-PM$_1$ mass concentrations for the high-ALW and low-ALW episodes were 100.8 μg m$^{-3}$
and 76.2 μg m$^{-3}$, respectively.

The relative daily increments of $frac_{org}$, $\kappa_{org}$, $\kappa_{org} \cdot frac_{org}$ (indicates the contribution of organics to $\kappa_{total}$), and $\kappa_{total}$ during the
classified 12 high-ALW haze episodes and 8 low-ALW haze episodes were averaged separately. Daily increments were used,
not hourly increments, to avoid the impact of diurnal variability. The first and last day of the episodes were not included in the
analysis as they were usually clean days, so that the chemical evolution was different from the hazy days. To minimize the
influence of transport or large local primary emissions, the relative daily increments of more than 40% were not included in
further analysis.
**3. RESULTS AND DISCUSSION**
**3.1 Aerosol liquid water contributed by organics**
The contribution of ALW$_{org}$ to ALW is the highest when NR-PM$_1$ mass concentrations are below 25 μg m$^{-3}$. In this low mass
loading, ALW$_{org}$/ALW varies widely between ~10% and ~80%, with an average of 32% (Fig. 1a). The high ALW$_{org}$/ALW in
low aerosol mass concentrations can be explained by high organics/NR-PM$_1$ mass fractions (57 ± 15%) (as shown in Fig. 1b)
and high $\kappa_{org}$ (as shown in Fig. 2). The striking variability in ALW$_{org}$/ALW is the result of highly variable chemical
compositions during clean days. In addition, higher uncertainties in NR-PM$_1$ measurements of low NR-PM$_1$ loadings and in
ALW modelling at low RH may also contribute to the large variability. High ALW$_{org}$/ALW in low aerosol mass concentrations
is consistent with previous studies (Dick et al., 2000; Fajardo et al., 2016). Those studies showed that modelled ALW$_{inorg}$ was
much lower than measured total ALW under low aerosol mass loadings in Beijing (Fajardo et al., 2016) and that ALW$_{org}$ was





comparable to $ALW_{inorg}$ in low RH (Dick et al., 2000).

As NR-PM$_1$ mass concentrations below 25 μg m$^{-3}$ increase to above 100 μg m$^{-3}$, $ALW_{org}$/ALW fraction decreases from an
average of 32% to 18% in Beijing (Fig. 1a). This decrease is mainly caused by the decrease of organics/NR-PM$_1$ mass fractions
from an average of 57% to 34% (Fig. 1b), and the decrease in organic/NR-PM$_1$ correlates with elevated RH, as indicated by
the color of the scattered points. Although organic concentration increases with rising RH and NR-PM$_1$, the concentration of
inorganic water-soluble salts increases even more, leading to a decreased fraction of organics. Variation of $ALW_{org}$/ALW
narrows as NR-PM$_1$ mass concentration increase. During high aerosol concentration, the aerosols are aged and dominated by
secondary species (Huang et al., 2014); while during low concentration, the origins of aerosol are more complex and variable.
As a result, the chemical composition of NR-PM$_1$ become more homogeneous with the increase in NR-PM$_1$.

$ALW_{org}$ calculated using the real-time $\kappa_{org}$ is much larger than that using a fixed $\kappa_{org}$ (0.08), which has often been used to
represent the hygroscopicity of urban organic aerosols (Nguyen et al., 2016). However, $\kappa_{org}$ in Beijing varies remarkably
between 0.06 and 0.26, with an average of 0.16 ± 0.04, much higher than 0.08. This higher $\kappa_{org}$ results in a higher $ALW_{org}$
fraction (18-32%) calculated in our study than predicted in previous ones (Nguyen et al., 2016; Wu et al., 2018). We note
higher $\kappa_{org}$ could be introduced via the conversion from organic O/C (Eq. 2); though $\kappa_{org}$ calculated from others
parameterizations (Chang et al., 2010; Duplissy et al., 2011; Peter et al., 2006; Raatikainen et al., 2010) are even higher than
from the one used here (Fig. S2). Also, based on previous reports that Q-ACSM can report higher $f_{44}$ values than the HR-ToF-
AMS (Fröhlich et al., 2015), there is a possibility that positive deviations of $f_{44}$ were introduced via the Q-ACSM measurements.
Despite these possibilities, the large variations in $\kappa_{org}$ emphasize the need to use real-time $\kappa_{org}$ instead of a fixed value. When
real-time $\kappa_{org}$ is not available, at least a localized average $\kappa_{org}$ should be considered.
**3.2 Influence of temperature, ALW, and NR-PM$_1$ mass concentrations on organic hygroscopicity**
Organic O/C ratio, and the derived organic hygroscopicity, increase with temperature ($T$) for all the four seasons (Fig. 2). This
positive correlation is more significant when $T$ is below 15 °C. For the different seasons, average O/C ratios for summer, spring,
autumn, and winter are 0.96, 0.82, 0.70, and 0.55, with corresponding average $T$ of 27.6, 14.6, 10.0, and 2.3 °C, respectively.
Diurnally, organic O/C show clear peaks at 14:00-16:00 which matches the diurnal variation of $T$ well (Fig. S3). Similar diurnal
changes of organic O/C have been previously observed (Hu et al., 2016; Sun et al., 2016). The promoting effect of $T$ upon O/C
can be attributed to multiple processes. On one hand, $T$ often correlates with higher solar radiation and atmospheric oxidative
capacity. On the other hand, higher $T$ accelerates gas phase and aqueous/heterogeneous phase reactions and thus increases O/C.
In addition, higher $T$ promotes the partitioning of semi-volatile species from particle phase to gas phase, also resulting in an
increase in O/C.




Fig. 3 shows the influence of ALW and NR-PM$_1$ mass concentration on organic O/C, or organic hygroscopicity. The cross-
impact of $T$ to O/C was separated by looking at the same color in Fig. 3. When ALW volume fraction is high (above 0.2-0.3),
organic O/C tends to increase with increasing ALW volume fraction; the increasing trend was most significant for spring and
autumn, while less significant for winter (Fig. 3a, c, d). The area between the two black lines in Fig. 3a, c, d is dominated by
the influence of ALW. Elevated ALW facilitates aqueous/heterogeneous reactions and promotes the formation of more oxidized
organics, such as dicarboxylic acids, thus increases O/C.

When ALW volume fraction is low (below 0.2-0.3), organic O/C decreases with lower NR-PM$_1$ mass concentration, indicated
by the size of the scattered points; this was observed in spring, autumn, and winter. One reason might be that at extremely low
aerosol mass concentrations, new particle formation events frequently occur and smaller particles dominate size distribution
(Cai et al., 2017; Guo et al., 2014). During formation and initial growth of new particles, extremely low volatile organic
compounds with the highest O/C ratio dominate; while subsequent growth involves organics with higher volatility and lower
O/C ratio (Donahue et al., 2013; Ehn et al., 2014). As a result, particle organic O/C decreases with growth of aerosol mass
concentration during new particle formation and growth events. Another possibility is that increased aerosol mass often
coincides with diminished solar radiation which suppresses photochemistry and may decrease organic O/C. In addition, a
fraction of the particles during clean periods are transported from less populated mountain areas. During such long-range
transport, atmospheric oxidation can increase O/C. Low ALW volume fraction correlates with low NR-PM$_1$ mass loadings,
which makes it look like organic O/C is decreasing with increasing ALW volume fraction. Overall, the apparent opposite trends
during high and low ALW volume fraction periods can actually be explained by a competition between the opposite impact of
ALW and NR-PM$_1$ mass loadings on organic evolution. However, summer was un exception, where no obvious dependence
of organic O/C on ALW volume fraction or NR-PM$_1$ mass concentration was observed.

The competing effects of ALW volume fractions and NR-PM$_1$ mass concentrations on organic O/C were further confirmed by
comparing organic evolution during the high and low-ALW haze episodes. Fig. 4 shows two typical haze episodes in Beijing,
with more chemical and meteorological information given in Fig. S5. During the high-ALW episode, where ALW contributes
0.2 - 0.75 to the total aerosol volume, organic O/C increases with haze accumulation. The increase of nighttime O/C is more
striking than that of daytime, likely due to the more abundant ALW at night (see Fig. S4). On the contrary, during the low-
ALW episode, where ALW volume fraction does not exceed 30%, daytime organic O/C decreases despite the increasing ALW
and $T$; this indicates that the decrease in O/C introduced by reduced photo-oxidation process and gas-particle partitioning is
larger than the O/C increase from aqueous/heterogeneous reactions. Nighttime O/C remains relatively constant, suggesting
that the promoting effect of aqueous/heterogeneous reactions on O/C is comparable to the reducing effects on O/C.





### 3.3 The influence of RH and particle hygroscopicity on particle hygroscopic volume growth factor

Particle volume growth factor increases rapidly with RH and particle hygroscopicity (Fig. 5). When RH is less than 80%, particle VGF increases slowly from 1 to 2.5 with rising RH; when RH exceeds 80%, VGF increases rapidly to above 5. This is generally consistent with previous studies (Bian et al., 2014). As shown in Fig. 5, significant variation of $\kappa_{total}$ also plays an important role in the change of water uptake. The dispersion of points in the vertical direction represents the influence of particle chemical compositions to ALW. For instance, when RH is fixed at 60%, VGF increases from 1.2 to 1.9 when $\kappa_{total}$ increases from ~0.20 to ~0.45.

The seasonal variations also reflect a combined promoting effect of RH and $\kappa_{total}$ on VGF. The average VGFs for spring, summer, autumn, and winter are 1.4, 1.6, 1.3, and 1.3, respectively. The highest VGF in summer is attributed to a combination of the higher frequency for high RH (red step line, compared to green, orange, and blue step line in Fig. 5b) and the relatively high particle hygroscopicity, $\kappa_{total}$ (0.35, compared to 0.38, 0.30, and 0.33 for other seasons).

A consequence of the high RH and high ALW is the higher particle overall hygroscopicity, $\kappa_{total}$, as compared with that at the low RH (Fig. 5). Aerosols are dominated by less hygroscopic particles ($\kappa_{total} < 0.3$) for RH below ~40% while aerosols are dominated by more hygroscopic particles ($\kappa_{total} > 0.4$) for RH above ~80% (Fig. 5). This suggests positive feedback between overall particle hygroscopicity and ALW. Higher $\kappa_{total}$ leads to higher ALW in similar RH while higher ALW, or higher RH, in turn corresponds to higher $\kappa_{total}$.

### 3.4 Interactions between organic evolution and particle hygroscopicity during high and low-haze episodes

During high-ALW episodes, the organic volume fraction decreases and organic hygroscopicity increases substantially during the accumulation of pollution. The average $frac_{org}$ is 0.51 and the daily increment of $frac_{org}$ is -11% (Fig. 6). The negative $frac_{org}$ increment indicates decreasing $frac_{org}$ which reflects the larger increase of inorganic soluble compounds (sulfate, nitrate, ammonium, and chloride) compared to that of organics during haze episodes. The average $\kappa_{org}$ is 0.165 and the relative daily increment of $\kappa_{org}$ is 8%. The positive $\kappa_{org}$ increment during high-ALW episodes reflects increasing $\kappa_{org}$ due to the effect of aqueous/heterogeneous reactions. To sum up, although the organic fraction decreases during the high-ALW haze episodes, the organic hygroscopicity increases. As a result, the contribution of $ALW_{org}$ to total ALW does not decrease as fast as the decrease of organic fraction.

During low-ALW episodes, the decrease in organic volume fraction is slower than that during high-ALW episodes, and organic





hygroscopicity hardly changes in the haze evolution process. The average $frac_{org}$ is 0.63 and the daily increment of $frac_{org}$ is -
4% (Fig. 6), of which both are higher than those in high-ALW episodes. This suggests that organic is still the dominating
component as haze accumulated during low-ALW episodes. The average $\kappa_{org}$ is 0.152 and the relative daily increment of $\kappa_{org}$ is
-1%, both of which are lower than those in high-ALW episodes. The near zero increment of $\kappa_{org}$ is a consequence of the
competition between aqueous/heterogeneous reactions and other processes. To sum up, the effects of ALW on chemical
compositions during low-ALW episodes are limited compared to high-ALW episodes.

As a consequence of the more significant changes in chemical composition during high-ALW episodes, the increase in particle
hygroscopicity is larger for high-ALW episodes than for low-ALW episodes. The relative daily increments of $frac_{org} \cdot \kappa_{org}$ during
high-ALW and low-ALW episodes are -4% and -3%, respectively (Fig. 6c). These negative increments indicate the negative
effect of the organic hygroscopic term on $\kappa_{total}$ during haze episodes. For high-ALW episodes, this means that the increase in
organic hygroscopicity in high-ALW episodes does not compensate for the effect of decreasing organic fraction. However, the
average daily increments of $\kappa_{total}$ during high-ALW and low-ALW haze episode are 8% and 2%, respectively (Fig. 6d). As $\kappa_{inorg}$
is fixed to 0.6 and the increment of $frac_{inorg}$ is opposite to that of $frac_{org}$, the positive $\kappa_{total}$ increment is a result of the positive
increment of the term $frac_{inorg} \cdot \kappa_{inorg}$.

The rapid decrease in $frac_{org}$ and increase in $\kappa_{org}$ during high-ALW episodes increase $\kappa_{total}$, which in turn promotes the ability
of particles to uptake water, forming positive feedbacks with ALW. The decrease of $frac_{org}$ or increase of $frac_{inorg}$ plays a
dominating role while the increase in $\kappa_{org}$ plays a minor but non-negligible role in increasing $\kappa_{total}$. During low-ALW episodes,
the positive feedbacks are weak or does not exist because both $frac_{org}$ and $\kappa_{org}$ do not change significantly.

There are other factors, not taken into consideration here, that might also affect ALW. These factors include the presence of
crustal material or trace metals, detailed particle size distributions, interactions between inorganic and organic compounds,
organic surfactants, and the particle phase state (Bian et al., 2014; Fountoukis and Nenes, 2007; Nakao, 2017; Ovadnevaite et
al., 2017). As a result, we strongly encourage that long term measurements of ALW and $\kappa_{org}$ be performed to test the results
shown here or establish a more reliable relationship between organic properties and ALW in the real atmosphere.

## 301   **4 Conclusion**

Our study emphasizes the need to include aerosol liquid water contributed by organics (ALW$_{org}$) in ALW modelling in Beijing,
instead of only using the inorganic contribution to total ALW. The reason is that ALW$_{org}$ contributes an average of 18-32% to
the total ALW in Beijing, according to our modelling results with ISORROPIA-II, κ-Köhler theory, and the ZSR mixing rule.





It is also necessary to use a real-time $\kappa_{org}$ to evaluate $ALW_{org}$, Since organic O/C, which has been shown in previous studies to
have a linear relationship with $\kappa_{org}$, varies from 0.2 to 1.3 in different seasons in Beijing. Using a fixed $\kappa_{org}$ (0.08) for typical
urban areas underestimates $ALW_{org}$ by a factor of ~2 in Beijing. When real-time $\kappa_{org}$ is not available, a localized average $\kappa_{org}$
should be used. O/C, or $\kappa_{org}$, generally increases with rising temperature and rising ALW in spring, autumn, and winter in
Beijing.

Positive feedback loops were found between $\kappa_{total}$ (which was determined by $frac_{org}$ and $\kappa_{org}$, as $\kappa_{inorg}$ was assumed to be 0.6)
and ALW during high-ALW episodes, with a conceptual diagram shown in Fig. S6. During high-ALW haze episodes, the
strong aqueous/heterogeneous phase reactions lead to a rapid decrease in $frac_{org}$ and increase in $\kappa_{org}$. These variations increase
$\kappa_{total}$, thus further promoting the uptake of water and forming positive feedbacks. These positive feedbacks were much weaker
in low-ALW episodes. The positive feedback loop between chemical composition evolution (mainly indicated by $frac_{org}$ and
$\kappa_{org}$) and ALW during high ALW-episodes is a driver for the severe haze episodes in Beijing.

## Acknowledgments

Financial support from the National Key R&D Program of China (2016YFC0200102) and the National Science Foundation of
China (91643201) is acknowledged.

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



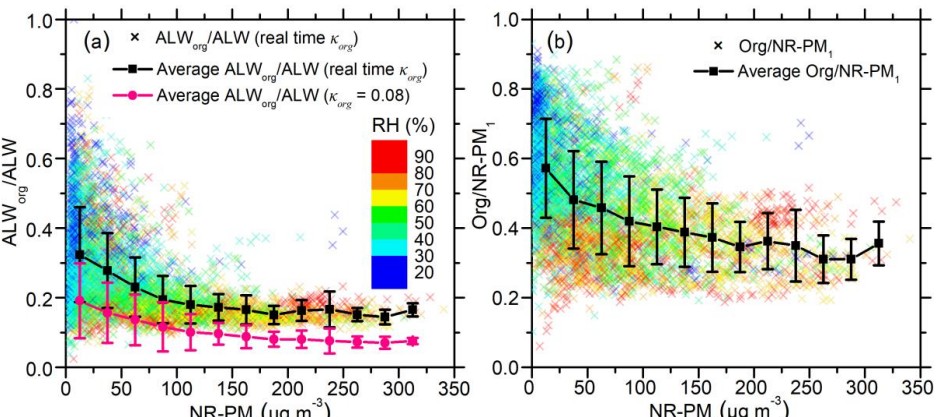


**Figure 1. (a) The colored scatter points represent the fraction of aerosol liquid water contributed by organics (ALWorg/ALW), which**

**was calculated using real-time $\kappa_{org}$. The black line shows the average of the colored points in each NR-PM₁ mass concentration bin.**

**The pink line is the average ALWorg/ALW calculated using a fixed $\kappa_{org}$ (0.08) in each NR-PM₁ mass concentration bin. (b) The colored**

**scatter points represent the organic mass fraction in non-refractory submicron aerosol (NR-PM₁). The black line is the average of**

**the colored points in each NR-PM₁ mass concentration bin. All the scattered points in both figures are colored with relative humidity**

**(RH).**


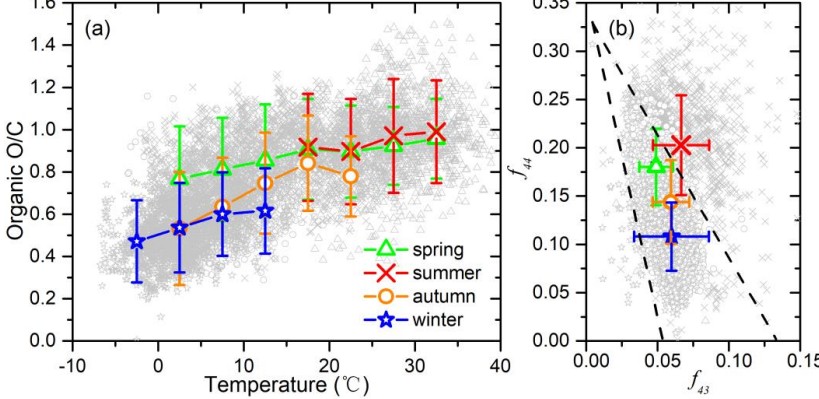


**Figure 2. (a) O/C ratio as a function of temperature in different seasons; (b) triangle plot ($f_{44}$ vs $f_{43}$) measured by the Q-ACSM in**

**different seasons**





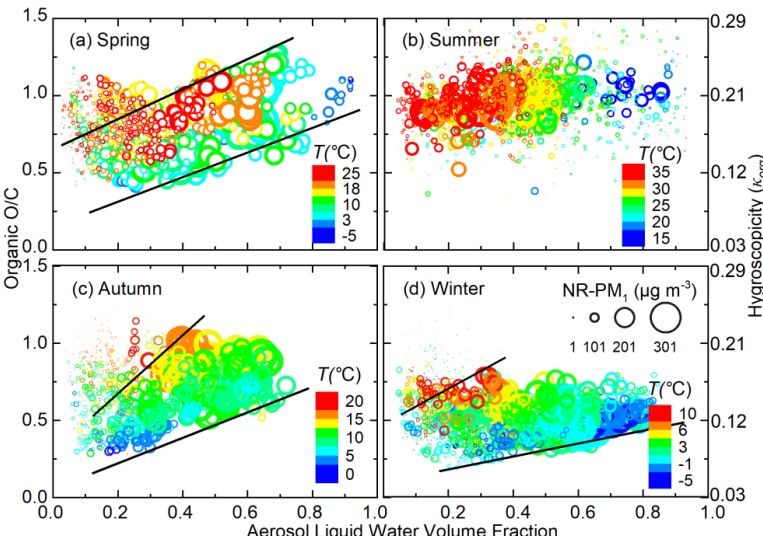


**Figure 3. Variation of organic O/C ratio (calculated from Q-ACSM measured $f_{44}$) as a function of aerosol liquid water (ALW) volume**

**fraction in different seasons. The size and color of the points represent the corresponding NR-PM$_1$ mass concentration and ambient**

**temperature, respectively. For spring, autumn, and winter, the areas between the two black lines represent the points less affected**

**by the gas-particle partitioning under low aerosol mass loadings.**


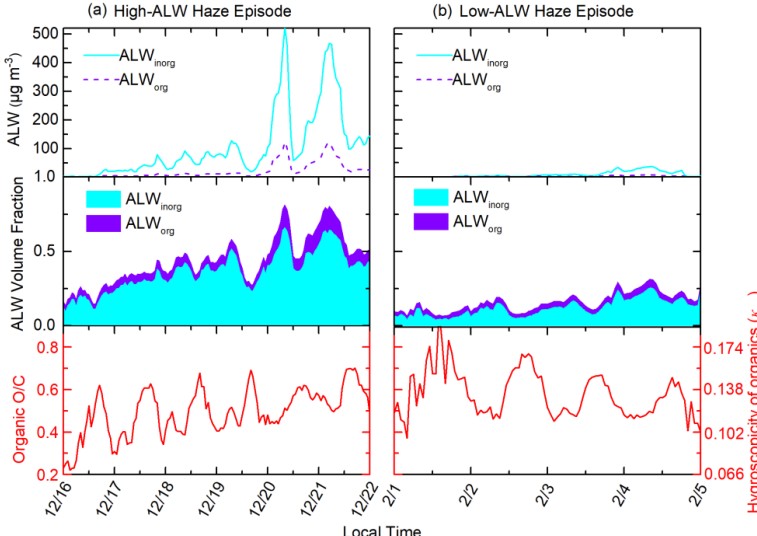


**Figure 4. Variations of aerosol liquid water contributed by organics (ALW$_{org}$), aerosol liquid water contributed by inorganics**

**(ALW$_{inorg}$), the volume fraction of total wet particle compositions, organic O/C during (a) a typical high-ALW episode and (b) a**

**typical low-ALW episode.**


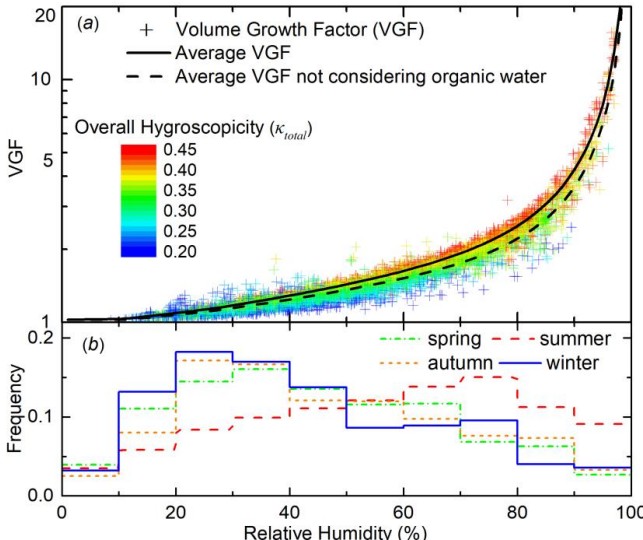


**Figure 5. (a) Volume growth factor (VGF, scattered points, calculated by Eq. 3) from the four seasons as a function of relative**

**humidity (RH). The points are colored by overall particle hygroscopicity ($\kappa_{total}$) calculated from aerosol bulk composition (Eq. 4).**

**The black line is the averaged VGF in different RH. Black dashed line is the average VGF without considering organic water. (b)**

**RH frequency during four seasons is expressed in step line.**

570

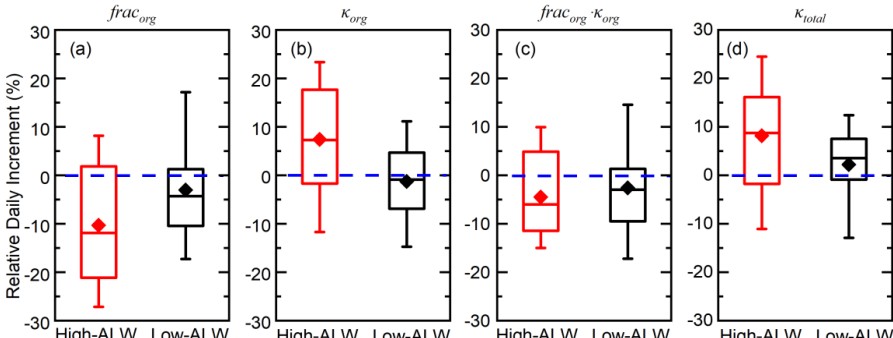

571

**Figure 6. Episode-based relative day increment of organic hygroscopicity ($\kappa_{org}$), organic volume fraction ($frac_{org}$), the hygroscopicity**

**term contributed by organics ($\kappa_{org}\cdot frac_{org}$), and overall particle hygroscopicity ($\kappa_{total}$) during high-ALW haze episodes and low-ALW**

**haze episodes. The box plots represent the 10th, 25th, 50th, 75th, and 90th percentiles of the corresponding data. The rhombus**

**represents the mean value of the corresponding data.**

576