# Peer review of "Interactions between aerosol organic components and liquid water content"

_Atmospheric Chemistry and Physics, 2019_

## Referee Comment (RC1) · Anonymous Referee #1 · 30 May 2019

Li et al. presented long-term measurements of chemical compositions for submicrometer particles in Beijing, used these measurement data to calculate aerosol liquid water (ALW), and discussed in feedback between ALW and aerosol chemical compositions. A novel aspect of this work is that the contribution of organics to ALW and the effect of chemical composition on hygroscopicity of organics were considered. This work could largely improve our understanding of formation and physiochemical properties of aerosol particles in Beijing. I would recommend it for final publication after the following comments are addressed.

**Scientific comments:**

Line 82-84: It is stated here that calculated ALW agreed well with measured ALW, as reported by previous work. However, no references are cited (the work by Zhang et al. (2007) only mentioned aerosol composition measurement, I presume). Proper references should be cited here, and preferably more quantitative results reported by previous work should be illustrated to support this statement.

My second comment is related to the first one. I agree that in principle it is better to use composition-dependent hygroscopicity, instead of fixed hygroscopicity, to calculate ALW associated with organic particles. However, it is not very clear to which extent the new approach would improve the agreement between measured and calculated ALW? Do the authors have access to measured ALW during some of these periods? A comparison between measurements and calculations using fixed and composition-dependent hygroscopicity should make this work more convincing.

Line 99: Please consider moving Figure S1 to the manuscript.

Line 194-195: please consider providing a figure which shows the frequency distribution of kappa values for organics.

Line 196-199: although kappa values calculated using different methods can be found in Fig. S2 and Table S2, it would be more convenient for readers if average values calculated using different methods can be stated in the main text.

Line 211: change "semi-volatile species" to "semi-volatile species (usually more oxidized)", because it seems to be non-apparent linkage between volatility and O:C ratios, at least for people who are not experts in organic particles.

Line 312: As Fig. S6 summarizes the concept proposed in this work, it may be worthy

moving it to the main text.

**Technical comments:**

Line 33: the perspective paper by Wu et al. (2018) should also be cited here.

The paper is very fluent; however, there are small grammatical errors which can be corrected. Perhaps the manuscript needs professional editing by a native speaker. Below I list some of the errors as examples:

Line 24: change "T" to "temperature (T)".

Line 24: change "…concentration. Among which…" to "…concentration, among which…"

Line 48: change "rising" to "rises".

Line 49: "transition" is a noun, please change it to a verb such as "change"

Line 74: change "specific" to "Specific".

**References:**

Wu, Z., Chen, J., Wang, Y., Zhu, Y., Liu, Y., Yao, B., Zhang, Y., and Hu, M.: Interactions between water vapor and atmospheric aerosols have key roles in air quality and climate change, Natl. Sci. Rev., 5, 452-454, 2018.

---

## Referee Comment (RC2) · Anonymous Referee #2 · 20 Jul 2019

This study reported the aerosol liquid water content contributed by organic components and its promotion on aerosol uptake. The results are very interesting. The manuscript is well-written and very clear. There are several questions should be addressed in the revised version.

(1) The ALW contributed by organics is calculated based on the relationship between oxidation state and korg. This means O:C values have very strong effect on the results and conclusions. The O:C values obtained from f44 using the parameterization method for Q-ACSM may highly variableïjĹCrenn et al., 2015ïjĿ. In addition, as given in Figure S2, different equations produced different korg. The relationship between O:C and korg

may vary with sampling locations. Therefore, the uncertainties on O:C calculation and the selection of equation for calculating korg should be discussed in the manuscript. (2) Line 211-212, the authors did give explanations why the partitioning of semi volatile species from particle phase to gas phase lead to an increase in O:C. (3) Line 141-145, the calculation of VGF assume the total volume equals to the addition of volumes of all components. If the irregular particles lead to a bias? What is the difference between VGF and mass-based growth factor? Why mass-based growth factor is not used here?

Crenn, V.; Sciare, J.; Croteau, P. L.; Verlhac, S.; Fröhlich, R.; Belis, C. A.; Aas, W.; Äijälä, M.; Alastuey, A.; Artiñano, B.; Baisnée, D.; Bonnaire, N.; Bressi, M.; Canagaratna, M.; Canonaco, F.; Carbone, C.; Cavalli, F.; Coz, E.; Cubison, M. J.; Esser-Gietl, J. K.; Green, D. C.; Gros, V.; Heikkinen, L.; Herrmann, H.; Lunder, C.; Minguillón, M. C.; Močnik, G.; O'Dowd, C. D.; Ovadnevaite, J.; Petit, J. E.; Petralia, E.; Poulain, L.; Priestman, M.; Riffault, V.; Ripoll, A.; Sarda-Estève, R.; Slowik, J. G.; Setyan, A.; Wiedensohler, A.; Baltensperger, U.; Prévôt, A. S. H.; Jayne, J. T.; Favez, O. ACTRIS ACSM intercomparison – Part 1: Reproducibility of concentration and fragment results from 13 individual Quadrupole Aerosol Chemical Speciation Monitors (Q-ACSM) and consistency with co-located instruments. Atmos. Meas. Tech. 2015, 8 (12), 5063-5087; DOI 10.5194/amt-8-5063-2015

---

## Author Comment (AC1) · 11 Aug 2019

**Responses to Reviewers' Comments on Manuscript ACPD-2019-316**

**(Interactions between aerosol organic components and liquid water content during haze episodes in Beijing)**

We are grateful for the two anonymous reviewers' comments that helped to improve this manuscript. We have addressed the comments in the following paragraphs and made corresponding changes in the revised manuscript. Comments are shown as *blue italic text* followed by our responses. Changes are highlighted in the revised manuscript and shown as underlined text in the responses. The references mentioned in the responses are listed at the end of this file.

**Reviewer #1:**

*Li et al. presented long-term measurements of chemical compositions for submicrometer particles in Beijing, used these measurement data to calculate aerosol liquid water (ALW), and discussed in feedback between ALW and aerosol chemical compositions. A novel aspect of this work is that the contribution of organics to ALW and the effect of chemical composition on hygroscopicity of organics were considered. This work could largely improve our understanding of formation and physiochemical properties of aerosol particles in Beijing. I would recommend it for final publication after the following comments are addressed.*

*Scientific Comments*

1) *Line 82-84: It is stated here that calculated ALW agreed well with measured ALW, as reported by previous work. However, no references are cited (the work by Zhang et al. (2007) only mentioned aerosol composition measurement, I presume). Proper references should be cited here, and preferably more quantitative results reported by previous work should be illustrated to support this statement.*

Response: Thanks for the suggestion. We add a reference here and show the quantitative results from that study. The relevant paragraph is revised as:

Line 82-84: ALW estimated by this method, which only requires aerosol chemical composition obtained from AMS measurements (Zhang et al., 2007), corresponds reasonably with measured ALW (The ratio of predicted ALW to measured ALW is 0.91, with $R^2 = 0.75$) (Guo et al., 2015).

2) *My second comment is related to the first one. I agree that in principle it is better to use composition-dependent hygroscopicity, instead of fixed hygroscopicity, to calculate ALW associated with organic particles. However, it is not very clear to which extent the new approach would improve the agreement between measured and calculated ALW? Do the authors have access to measured ALW during some of these periods? A comparison between measurements and calculations using fixed and composition-dependent hygroscopicity should make this work more convincing.*

Response: Thanks for the suggestion. We agree that a comparison with real measurement would make the calculation more convincing, but we didn't have access to any of the ALW measurement during our sampling period. Theoretically, this method should work based on many studies related with the organic hygroscopicity as stated in the main text. We cannot evaluate the accurate uncertainty of the method, but the trend of $ALW_{org}$ should be correct. An interesting finding of this study is that the increase of organic O/C (varies largely

between 0.2 and 1.3) during haze evolution can further promote the uptake of water. This finding relies more on the trend of $ALW_{org}$ instead of the absolute value of $ALW_{org}$. In the "Results and Discussion" part, we recommend future studies with measured ALW to do the comparison.

Line 300-301: As a result, we suggest that long term measurements of ALW and $\kappa_{org}$ should be performed to test the results shown here and to establish a more reliable and accurate relationship between organic properties and ALW in the real atmosphere.

*3) Line 99: Please consider moving Figure S1 to the manuscript.*

Response: Our manuscript mainly focuses on the interactions between organic hygroscopicity and ALW. Figure S1 is only a support for the uncertainty of calculation method for $ALW_{inorg}$. In addition, similar statements with Figure S1 have already been discussed in previous studies, e.g., Song at al (2018).

*4) Line 194-195: please consider providing a figure which shows the frequency distribution of kappa values for organics.*

Response: The frequency distribution of kappa values for organics used in our study is as following (calculated using the method from Lambe et al., (2011)). This information has been included in Figure S2, where the 10th, 25th, 50th, 75th, 90th percentiles, and the mean value of the corresponding $\kappa_{org}$ in each season were shown.

[Figure]

*5) Line 196-199: although kappa values calculated using different methods can be found in Fig. S2 and Table S2, it would be more convenient for readers if average values calculated using different methods can be stated in the main text.*

Response: Thanks for the suggestion. We add a sentence in the main text:

Line 200-201: As shown in Table. S2, the average $\kappa_{org}$ calculated from other methods are $0.22 \pm 0.07$ (Chang et al., 2010), $0.19 \pm 0.06$ (Massoli et al., 2010), and $0.21 \pm 0.12$ (Duplissy et al., 2011; Jimenez et al., 2009).

*6) Line 211: change "semi-volatile species" to "semi-volatile species (usually more oxidized)", because it seems to be non-apparent linkage between volatility and O:C ratios, at least for people who are not experts in organic particles.*

Response: We agree. The sentence is revised as following:

Line 214-215: In addition, higher $T$ promotes the partitioning of semi-volatile species (usually less oxidized than low-volatile species) from particle phase to gas phase, also resulting in an increase in particle organic O/C.

*7) Line 312: As Fig. S6 summarizes the concept proposed in this work, it may be worthy moving it to the main text.*

Response: Thanks for the suggestion. Figure S6 was moved to the main text as Figure 7.

[Figure]

**Figure 7. The positive feedback loops between aerosol liquid water and organic evolution during high-ALW haze episode in Beijing.**

*Technical comments*
*1)Line 33: the perspective paper by Wu et al. (2018) should also be cited here.*

Response: Thanks. The paper was added to the citations.

*The paper is very fluent; however, there are small grammatical errors which can be corrected. Perhaps the manuscript needs professional editing by a native speaker. Below I list some of the errors as examples:*

Response: Thanks for the suggestion. In the revised manuscript, we carefully checked the grammatical errors. It was also edited by a native speaker.

*2)Line 24: change "T" to "temperature (T)".*
*3)Line 24: change "…concentration. Among which…" to "…concentration, among which…"*
*4)Line 48: change "rising" to "rises".*
*5)Line 49: "transition" is a noun, please change it to a verb such as "change".*
*6)Line 74: change "specific" to "Specific".*

Response: Thanks, all of these are corrected in the revised manuscript.

**Reviewer #2:**

*This study reported the aerosol liquid water content contributed by organic components and its promotion on aerosol uptake. The results are very interesting. The manuscript is well-written and very clear. There are several questions should be addressed in the revised version.*

*Major Comments:*

*1.  The ALW contributed by organics is calculated based on the relationship between oxidation state and korg. This means O:C values have very strong effect on the results and conclusions. The O:C values obtained from f44 using the parameterization method for Q-ACSM may highly variableïjʹLCrenn et al., 2015ïjLʹ. In addition, as given in Figure S2, different equations produced different korg. The relationship between O:C and korg C1 may vary with sampling locations. Therefore, the uncertainties on O:C calculation and the selection of equation for calculating korg should be discussed in the manuscript.*

Response: Thanks, we agree that large uncertainties may be introduced during the Q-ACSM measurement, the conversion of $f_{44}$ to O/C, and the conversion of O/C to $\kappa_{org}$. Although we cannot quantitively evaluate the uncertainties, we believe the trend of $\kappa_{org}$ we obtained in the study is reliable given the numerous organic hygroscopicity studies. We modified the paragraph that describes the uncertainty as following:

Line 196-203: We note higher $\kappa_{org}$ could be introduced via the conversion from organic O/C (Eq. 2); though $\kappa_{org}$ calculated from others parameterizations (Chang et al., 2010; Duplissy et al., 2011; Massoli et al., 2010; Peter et al., 2006; Raatikainen et al., 2010) are even higher than from the one used here (Fig. S2). The average $\kappa_{org}$ calculated from other methods were 0.22 ± 0.07 (Chang et al., 2010), 0.19 ± 0.06 (Massoli et al., 2010), and 0.21 ± 0.08 (Duplissy et al., 2011; Jimenez et al., 2009) (Table S2). Also, based on previous reports that Q-ACSM can report higher $f_{44}$ values than the HR-ToF-AMS (Fröhlich et al., 2015) and that $f_{44}$ reported by Q-ACSM may be highly variable between different instruments (Crenn et al., 2015), there is a possibility that positive deviations and large uncertainties of $f_{44}$ were introduced via the Q-ACSM measurements.

*2.  Line 211-212, the authors didn't give explanations why the partitioning of semi volatile species from particle phase to gas phase lead to an increase in O:C.*

Response: Compared to low volatile species which tend to remain in particle phase in high ambient temperature, semi-volatile species usually correspond with low oxidation states and O/C ratio (Donahue et al., 2011). As a result, the repartitioning of less volatile species (with low O/C ratio) back to the gas phase may lead to an increase of O/C in particle phase. The sentence was revised as:

Line 214-215: In addition, higher $T$ promotes the partitioning of semi-volatile species (usually less oxidized than low-volatile species) from particle phase to gas phase, also resulting in an increase in particle organic O/C.

*3.  Line 141-145, the calculation of VGF assume the total volume equals to the addition of volumes of all components. If the irregular particles lead to a bias? What is the difference between VGF and mass-based growth factor? Why mass-based growth factor is not used here?*

Response: The irregular particles will lead to a bias in the calculation of size growth factor, but will not lead to a bias in VGF unless the density of the mixture is significantly different from the weighted average density of each species. The conversion to VGF is for better comparison with previous studies as VGF is more oftenly used in literature (e.g., Bian et al., 2014; Fajardo et al., 2016; Stanier et al., 2004) than mass-based growth

factor. What's more, the rates of aqueous processes are more directly related with volume concentration than mass concentration. The mass-based growth factor will be lower than VGF because the density of water is smaller than other particle species, but the trend should be in consistent with VGF.

**References**

[revised manuscript text omitted]